# Introducing Semi-Interpenetrating Networks of Chitosan and Ammonium-Quaternary Polymers for the Effective Removal of Waterborne Pathogens from Wastewaters

**DOI:** 10.3390/polym15051091

**Published:** 2023-02-22

**Authors:** Iulia E. Neblea, Anita-L. Chiriac, Anamaria Zaharia, Andrei Sarbu, Mircea Teodorescu, Andreea Miron, Lisa Paruch, Adam M. Paruch, Andreea G. Olaru, Tanta-V. Iordache

**Affiliations:** 1Advanced Polymer Materials and Polymer Recycling Group, National Institute for Research & Development in Chemistry and Petrochemistry ICECHIM, Splaiul Independentei No. 202, 060021 Bucharest, Romania; 2Department of Bioresources and Polymer Science, Faculty of Chemical Engineering and Biotechnologies, University “Politehnica” of Bucharest, 1–7 Gh. Polizu Street, 011061 Bucharest, Romania; 3Norwegian Institute of Bioeconomy Research (NIBIO), Division of Environment and Natural Resources, Oluf Thesens vei 43, 1433 Aas, Norway; 4S.C. EDAS-EXIM S.R.L., Banat Street 23, 010933 Bucharest, Romania

**Keywords:** semi-interpenetrating hydrogels, chitosan, quaternary-ammonium monomers wastewater treatment, bacteria and pathogens removal

## Abstract

The present work aims to study the influence of ammonium-quaternary monomers and chitosan, obtained from different sources, upon the effect of semi-interpenetrating polymer network (semi-IPN) hydrogels upon the removal of waterborne pathogens and bacteria from wastewater. To this end, the study was focused on using vinyl benzyl trimethylammonium chloride (VBTAC), a water-soluble monomer with known antibacterial properties, and mineral-enriched chitosan extracted from shrimp shells, to prepare the semi-IPNs. By using chitosan, which still contains the native minerals (mainly calcium carbonate), the study intends to justify that the stability and efficiency of the semi-IPN bactericidal devices can be modified and better improved. The new semi-IPNs were characterized for composition, thermal stability and morphology using well-known methods. Swelling degree (SD%) and the bactericidal effect assessed using molecular methods revealed that hydrogels made of chitosan derived from shrimp shell demonstrated the most competitive and promising potential for wastewater (WW) treatment.

## 1. Introduction

The rapid development of various industry fields is posing increasing pressure on the management of contaminant effluents, which can be discharged directly into main surface water bodies [1]. The wastewaters of many industries such as food, paint, textile, pharmaceuticals, or battery manufacturers may contain large quantities of human pathogenic bacteria or viruses, metal ions, organic solvents, dyes or other materials, which can be released into the receiving water bodies and eventually enter the food chain, posing high risks to humans and the environment [2,3,4]. Among many proven water treatment methods such as coagulation and flocculation [5], membrane separation [6], biological treatment [7] or reverse osmosis [8], and most recently, solar steam generation [9,10], the adsorption process is an often-applied method due to its distinct effectiveness and economic advantages. This process involves either chemical adsorption through an irreversible process, or physical adsorption through a process controlled by physical interactions [11]. In this regard, there are many scientific publications reporting the use of polymeric materials such as membranes or hydrogels as absorbents for water pollutants [12,13,14,15,16].

Due to their tunable properties and hydrophilic nature, hydrogels have been popularly used as various tools in medical care and pharmacy, such as drug-delivery for controlled release or as various grafts, implants [17], wound bandages and patches for the controlled release of drugs [18]. They have also been used as materials for artificial tendons, membranes, and artificial articular cartilage or artificial skin [19]. These 3D insoluble materials, which can absorb large amounts of liquids, have been described in several publications, elucidating their antibacterial potential in the biomedical field, for wastewater treatment by removing bacteria, dyes and other compounds from contaminated waters [20]. Hydrogels can be obtained from homopolymer formulations, copolymer formulations or semi-interpenetrating polymer networks (semi-IPNs) [21]. Given the fact that homopolymer networks alone rarely present the desired physical-chemical properties, multicomponent 3D networks such as semi-IPNs have been intensively studied due to the possibility of combining the favorable properties of each polymeric component into a new system with improved or even new features. IPNs and semi-IPNs are an important class of materials, consisting of two types of polymers that intertwine without any chemical interactions between the two [22]. Such 3D structures are usually obtained by either the simultaneous polymerization of two non-copolymerizing monomers using two different polymerization/crosslinking mechanisms (e.g., polycondensation [23] or polyaddition [24] and radical polymerization [25]) to form double crosslinked interpenetrating networks of polymers (IPNs), or by polymerization/crosslinking a monomer or a pre-polymer around an existing polymer chain [26,27] to form semi-IPNs.

Currently, some popular topics in the field of water treatment include the use of polysaccharides (e.g., alginate [28], chitosan [29], chitin [30] or cellulose [31]) due to their interesting properties such as low toxicity and price, biocompatibility, and biodegradability. In this regard, chitosan is a widely used polysaccharide for obtaining semi-IPN polymeric hydrogels, with its remarkable biological properties, such as biodegradability, biocompatibility, and antibacterial activity [32]. Chitosan is a linear cationic biopolymer, which can be obtained by the N-deacetylation of chitin from crustacean carcasses, insect exoskeletons or fungal cell walls [33]. Due to the high content of amino and hydroxyl functional groups in its composition, chitosan has attracted much attention for its use as an absorbent material, which has a high affinity for proteins, dyes, and metal ions [34,35]. Additionally, the presence of the amino (–NH_2_) and hydroxyl (–OH) reactive groups in this structure has made possible the graft polymerization of hydrophilic vinyl monomers [36]. In order to develop new and more efficient controlled release systems of the incorporated active substance, numerous studies have been conducted on the production of IPN or semi-IPN polymers based on chitosan or its derivatives, as well as other polysaccharides or their derivatives [37]. Moreover, chitosan is soluble in an acidic medium in the presence of amine groups from the 56 D-glucosamine units [38]. It can be crosslinked by increasing the pH of the solution or by casting the solution in a non-solvent, which is ideal for wastewater treatment and heavy metal retention, as described by Bhullar et al. [39] in their paper regarding semi-IPNs of chitosan/acrylic acid and thiourea hydrogels. The possibility of using chitosan-based materials for environmental applications was also explored by Wan et al. [40], by obtaining zirconium-loaded chitosan/poly(vinyl alcohol) IPN hydrogels for phosphate sorption. The antibacterial properties were demonstrated by Mohamed et al. [41] using semi-IPN hydrogels based on carboxymethyl chitosan and poly(acrylonitrile).

More recent studies have also reported the use of quaternary ammonium salts as biocidal agents in medical applications and wastewater treatment [42,43,44]. Compounds with quaternized amines such as pyridinium, triethylammonium bromide or vinyl benzyl trimethylammonium chloride are well known to possess antimicrobial activity against specific microorganisms such as *Staphylococcus aureus*, *Escherichia coli*, *Candida albicans*, *Vibrio fischeri* and *Staphylococcus albus* [45]. Due to their low toxicity, these compounds have been used as solutions for open wounds and as preoperative disinfectants [46,47]. However, the high cost of quaternary ammonium salts, particularly for water purification, has driven the research on more cost-effective routes and, therefore, several studies have reported the preparation of quaternary-ammonium-grafted polymers to increase antibacterial performance and combine the intrinsic properties of the corresponding polymers. Among them, some studies have reported the chemical grafting of ammonium salts along chitosan macromolecular chains, which resulted in improved natural antibacterial and antifungal properties [48].

Although the application of quaternary ammonium salts as bactericidal agents is still a challenge in this field, studies have reported its effective action against Gram-negative bacteria such as *Escherichia coli* [42,49,50,51]. Considering this, the current study illustrates the bactericidal potential of a novel semi-IPN hydrogel for wastewater treatment. In this regard, vinyl benzyl trimethylammonium chloride (VBTAC), a water-soluble cationic monomer with antibacterial properties, and mineral-enriched chitosan extracted from shrimp shells are used to prepare the semi-IPNs. The reason for using mineral-enriched chitosan (i.e., chitosan obtained by passing the demineralization stage [52]) lies in the intention of improving the stability for chitosan-based hydrogels using the native minerals present in its structure (mainly calcium carbonate). Plus, the process of producing this type of chitosan and, thereafter, the semi-IPN hydrogels is also remarkably cost-effective and environmentally friendly, with regard to the carbon footprint of the energy-consuming demineralization process [53]. The main features of this new material are: (i) an enhanced bactericidal effect supported by the high density of ammonia functional groups (-N^+^(CH_3_)_3_^−^Cl) as pendant groups on poly(VBTAC), coupled with amino functional groups of chitosan; (ii) an enhanced hydrophilicity and also affinity for organic residues upon the presence of poly(VBTAC), and last but not least, (iii) biodegradability conferred by the chitosan matrix but also by poly(VBTAC), which is endowed with environmentally friendly properties [54]. To demonstrate the performance of these new semi-IPN hydrogels based on mineral-enriched chitosan, comparison experiments were performed using commercial chitosan and chitosan obtained from commercial chitin.

## 2. Materials and Methods

### 2.1. Materials for the Synthesis of Semi-IPN Hydrogels

In order to obtain semi-IPN hydrogels with antibacterial properties, three types of chitosan were used: commercial chitosan (CC, Sigma-Aldrich St Louis, MO, USA, ≥75% deacetylation degree, Mn = 2.056 × 10^5^ g/mol), chitosan synthesized in previous works from commercial chitin via a deacetylation process (CCH, 77% deacetylation degree, Mn = 4.7291 × 10^5^ g/mol [52]) and mineral-enriched chitosan extracted in previous works from shrimp shells (SHC, 76% deacetylation degree, Mn = 9.058 × 10^3^ g/mol, crystallinity 59.42% on (020) plane [52]). Vinyl benzyl trimethylammonium chloride (Sigma-Aldrich, 99%, VBTAC) was used as a monomer to generate quaternary ammonium pendant groups. The synthesis of semi-IPN hydrogels was performed following the initiation process with 4,4′-azobis-4-cyanovaleric acid (Sigma-Aldrich ≥ 98%, ACVA) in the presence of N,N′-methylenebisacrylamide (Sigma Aldrich ≥ 98%, MBA) as a crosslinking agent. The listed compounds were used without any purification. Acetic acid (Sigma-Aldrich, 99%) and pure demineralized water were also used to solubilize chitosan.

### 2.2. Synthesis and Purification of Semi-IPN Hydrogels

CCH and SHC were prepared earlier, starting from either the commercial chitin or shrimp shells, respectively, and described by Miron et al. [51]. In brief, chitosan synthesis starting from shrimp shells involved only deproteinization and deacetylation, compared to the classical procedure which also included a demineralization process. Shrimp shells were first cleaned with water, desiccated with NaCl in direct sun light, and lyophilized. The deproteinization of dried shrimp shells was performed with an aqueous solution of 1 M NaOH (SH:NaOH = 1:15, *w*/*v*) at room temperature, under magnetic stirring (200 rpm) for 24 h, in order to obtain a mineral-enriched chitin. In the following step, the deacetylation of chitin was performed using a NaOH 50% solution (chitin:NaOH = 1:20, *w*/*v*), resulting in a mineral-enriched chitosan (SHC). The reaction was conducted in one batch for 6 h, at 80 °C, under magnetic stirring (400 rpm). After each procedure, the samples were washed with distilled water until neutral pH, and then filtered and lyophilized. CCH was synthesized using the same deacetylation protocol described previously for SHC, but using commercial chitin instead.

The synthesis of semi-IPN hydrogels based on VBTAC and chitosan was performed using free radical polymerization. In a typical batch, an aqueous solution consisting of VBTAC and MBA (1.25 wt.% rel. to monomer content) was first obtained, to which the ACVA (0.25 wt.% rel. to monomer content) initiator solution was added. A separate 10 mL solution of 1 wt.% chitosan (CC, CCH or SHC) was prepared at room temperature using a mixture of water and acetic acid. The semi-IPN hydrogels were obtained by mixing the first prepared solution and 1 mL of 1 wt.% chitosan solution into a glass vial, which was then sealed with a plastic cap, thoroughly shaken, and introduced into a water bath at 60 °C without steering. After 20 h, the vials were broken, and the obtained hydrogel was cut into pieces that were 5 mm in height. Since one of the goals of this study was to optimize the recipe for preparing efficient but also stable semi-IPN hydrogels, the chitosan/VBTAC ratio was varied for all the three series of samples: CC-based semi-IPNs, CCH-based semi-IPNs and SHC-based semi-IPNs. Table 1 summarizes the exact recipes applied when preparing the three series of hydrogels, denoted in accordance with their base materials, as follows: CC-IPNn, CCH-IPNn and SHC-IPNn, in which case n = 1–4. The sample denoted as polyVBTAC is a reference sample prepared in the same conditions as the rest of the hydrogels but without chitosan. Although it was expected that semi-IPNs without VBTAC would be unsuccessful, the study also included preparing CC-1, CCH-1 and SHC-1 as reference samples without VBTAC. Yet, the latter gels were unstable, being sticky and gummy, which forced us to exclude them from further studies.

The purification step involved introducing the samples into glass vials with 100 mL of distilled water and keeping them for 7 days at room temperature. The water was changed daily to remove the unreacted monomer. At the end, the swollen hydrogels were dried in the oven at 40 °C until constant weight (48 h).

### 2.3. Characterization Methods for Semi-IPN Hydrogels

Fourier-transform infrared spectrometry (FTIR) spectra of the samples were recorded on a Nicolet™ Summit PRO FTIR Spectrometer (Thermo Fisher Scientific, Waltham, MA, USA) using 16 scans in the range 400–4000 cm^−1^ with a resolution of 4 cm^−1^, on KBr pellets.

Thermo-gravimetric analysis (TGA) and derivative thermogravimetry (DTG) were assessed using TA Instruments Q5000 IR equipment by heating a sample of approx. 5 mg in the 25–700 °C temperature range, with a constant heating rate of 10 °C min^−1^. The thermal stability of the prepared hydrogels was investigated in order to underline the effect of different concentrations of VBTAC upon the semi-IPN networks, as complementary results to the FTIR measurements.

Scanning electron microscopy (SEM) images of the semi-IPN hydrogels were obtained using a Hitachi TM4000 plus II scanning electron microscope equipped with a secondary electron (SE) detector at an acceleration voltage of 15 kV.

### 2.4. Swelling Degree (SD) Determination for the Semi-IPN Hydrogels

The main characteristic of the hydrogels is the swelling capacity of the polymer network represented by the amount of liquid retained by the material. The swelling degree study was conducted considering the gravimetric method. Thus, the swelling degree was determined by introducing a weighed xerogel disc (*m_dry_*) into distilled water at room temperature. The swollen samples were removed from the liquid at certain times, the excess water was wiped with filter paper, and the mass of the hydrogel (*m_wet_*) was weighed and reintroduced in the liquid. The assessment was performed until the swollen samples reached a constant weight, which corresponds to equilibrium, and the SD was determined using Equation (1).
(1)$$SD=\frac{m_{wet}-m_{dry}}{m_{dry}}\;(g\;\mathrm{water}/g\;\mathrm{dry}\;\mathrm{polymer})$$

### 2.5. Wastewater Sampling and Bacteriological Testing of Semi-IPN Hydrogels

The industrial wastewater (WW) samples were collected from the sewage water (SW) source directly from the homogenization basin, in several stages depending on the factory’s operating program and its maintenance period. The WW samples were vacuum filtered using PES-membrane filters (PALL 516-0427, Φ47 mm, pore size 0.45 μm) and a Labbox filtration system. The filtered WW was used to determine the chemical indicators and the PES-membrane filters were used for bacteriological indicators; the values were compared with those obtained for WW alone (as a control sample).

All measurements were performed in static isothermal conditions and the samples were analyzed in triplicate. For a classic measurement, approx. 0.5 g of the hydrogel sample was placed into a glass flask over which 150 mL of WW was added. The vessel was sealed and left at room temperature (24 °C) away from light sources. After 24 h, the WW supernatant from the sample was collected and filtered using conventional membrane filtration methods, and the swollen hydrogel structures were placed in a clean vessel.

The semi-IPN materials were analyzed before and after direct contact with the WW samples from the SW source. After 24 h of contact time, the average number of coliform bacteria and clostridia was determined for each sample using the membrane filtration method and via comparison with a control sample. The bacteriological analyses included firstly the placement of the PES-membrane filters on agar plates, so that no air was trapped between the filter and the medium. The incubation took place in a Hach Incubator at 44 ± 0.5 °C for 24 h, where the filter disks were transferred to chromogenic/TSC agar plates. The results were observed approx. 15 min after removing the filters from the incubator. The bacteria colonies were counted using a Funke Gerber colony counter.

### 2.6. Genetic-Marker-Based Quantitative PCR (qPCR) Analyses of Waterborne Pathogens

#### 2.6.1. Genomic DNA Extraction of WW Materials

The PES-membrane filters resulting from the WW filtrations (residues and filters) underwent DNA extraction. Genomic DNA was extracted from each replicate (3 replicates per WW sample) of each treated and untreated WW sample using a DNeasy PowerWater kit (Qiagen GmbH, Hilden, Germany). The yielded DNA concentration and integrity were measured on a mySPEC Spectrophotometer (VWR, Radnor, PA, USA). The final concentration of purified DNA ranged from 20 to 45 ng·μL^−1^ with both 260/280 and 260/230 ratios between 1.8 and 2.0.

#### 2.6.2. Molecular Detection and Quantification of Microbial Pathogens in WW

Common waterborne bacterial pathogens, such as *Campylobacter jejuni* (*C. jejuni*), *Enterococcus faecalis* (*E. faecalis*), *Salmonella enterica* serovar Typhimurium (*S. Typhimurium*), *Clostridium perfringens* (*C. perfringens*), *Legionella pneumophila* (*L. pneumophila*), *Shigella* spp. and Shiga-toxin producing *E. coli* (STEC), were molecularly examined using the developed panel of species-specific genetic markers [42,55]. Additionally, protozoan pathogens such as *Cryptosporidium parvum* (*C. parvum*) and *Giardia lamblia* (*G. lamblia*) were analyzed in each sample using the established genetic markers [56]. Pathogen qPCR assays were performed in duplicate on a Bio-Rad CFX Connect Real-time PCR Detection System (Irvine, CA, USA). In a 20 µL total qPCR reaction, 10 μL of SsoAdvanced™ Universal Probes Supermix was mixed with 500 nM of each primer, 250 nM 5′-FAM probe and sterile nuclease-free H_2_O. For the initial denaturation step, the reaction was heated at 95 °C for 3 min, followed by 40 cycles at 95 °C for 15 s and 60 °C for 30 s. The standard curve was established using 10-fold serial dilutions of the target gene-carrying plasmids (from 106 to 100 copies·μL^−1^). The quantity of detected pathogen was analyzed using CFX Manager Version 3.1 (Bio-Rad, Irvine, CA, USA). The pathogen reduction rate for each treatment was deduced from the quantity comparison of the copy numbers of the target genetic marker carried in treated vs. untreated WW samples.

## 3. Results and Discussion

### 3.1. FTIR Spectroscopy of Semi-IPN Hydrogels

The FTIR spectra represented in Figure 1a–c indicate the characteristics of the synthesized chitosan-based semi-IPN hydrogels. The sample sets are distinguished primarily by the presence of the stretching vibration band characteristic of the -NH bond in amide II from the chitosan structure at 1562 cm^−1^ [57]. The samples may also indicate the presence of chitosan through the bending vibration band corresponding to the C=O bond of amide I observed at 1654 cm^−1^, but in which case might overlap with other peaks characteristic of VBTAC’s presence. Moreover, the vibration of -OH groups and C-H bonds can be observed at 3410 cm^−1^ and 2945 cm^−1^, respectively. In addition to the bands specific to chitosan, the samples show bands at 1489 cm^−1^ and at 960 cm^−1^, indicating the quaternary ammonium group vibration, C-N^+^(CH_3_)_3_, specific for the synthetic polymer VBTAC, and a peak at 3020 cm^−1^ specific to the C-H vibration from the macromolecular chain of poly(VBTAC) [58] (Figure 1a–c). The specific values for each representative peak are listed in Appendix A.

### 3.2. Thermo-Gravimetric Analysis (TGA/DTG) for Synthesized Semi-IPN Hydrogels

The thermal stability of the semi-IPN hydrogels was investigated using TGA/DTG analysis. Figure 2A,B and Figure 3A,B represent the thermograms of IPN hydrogels containing CCH and CC. The samples show similar thermal degradation behavior to the samples containing SHC (Figure 4A,B), with two main degradation stages indicating the decomposition of quaternary ammonium groups in the polyVBTAC structure. The second stage was characteristic of the chitosan-polyVBTAC network cleavage, although the degradation of chitosan might overlap with the first degradation stage of polyVBTAC. Moreover, the DTG curves represented in Figure 2B and Figure 3B indicate first the dehydration process at ~96 °C, followed by a first degradation stage between 250 and 350 °C (first peak at ~266 °C indicating the degradation of quaternized ammonium groups in the polyVBTAC and the second peak at ~316 °C characteristic of chitosan degradation) and a second stage at a maximum of 429 °C indicating the degradation of the IPN polymer network. The specific degradation temperatures and the weight loss of each semi-IPN hydrogel is visualized in Appendix A.

Furthermore, Figure 4A,B indicates the thermograms of the synthesized hydrogels based on SHC and polyVBTAC compared with SHC and polyVBTAC alone. The TGA curves of the analyzed samples show a similar thermal degradation profile, indicating two degradation stages, the first being characteristic of the thermal decomposition of quaternary ammonium groups in the polyVBTAC structure and the second stage being characteristic of the chitosan–polyVBTAC interpenetrating network cleavage [58]. As shown in Figure 4A, there are no significant differences in weight loss, about 80.5% for all the SHC-based hydrogels in the series. Yet, a significant mass loss of 84% is recorded for polyVBTAC alone. This difference can be explained by the fact that polyVBTAC led to the formation of a simple crosslinked polymer structure that degrades faster than the SHC/polyVBTAC semi-IPN, in which case the high amounts of calcium carbonate enhance the thermal stability, as was observed for SHC as well (only 55% mass loss).

As shown in Figure 4B, from the DTG curves of the SHC-based hydrogels, a first stage of thermal degradation with a decomposition temperature of 90 °C was registered for the water loss and a pronounced peak was observed at 429 °C due to the degradation of the polyVBTAC–SHC interpenetrated network. The other two peaks in the 250–350 °C temperature range indicate the fragmentation of quaternized ammonium groups from polyVBTAC and chitosan degradation, respectively. One interesting behavior can be noted in this series regarding these two latter peaks. The maximum decomposition temperature for quaternary ammonium groups increased with the increasing amount of VBTAC, in the following sequence: SHC-IPN_2_ (264 °C) < SHC-IPN_3_ (267 °C) < SHC-IPN_4_ (276 °C, hump). For polyVBTAC alone, only the peak indicating the degradation of quaternized ammonium groups at 295 °C and the degradation polymer network at a maximum of 406 °C can be observed. Thus, TGA analysis confirmed the FTIR analysis regarding the formation of semi-IPN hydrogels and highlighted the influence of VBTAC concentration on hydrogels, which has led to an increase in the thermostability of the semi-IPN materials at higher concentrations of VBTAC monomer.

### 3.3. Morphology of the Synthesized Semi-IPN Hydrogels

The micrographs of the lyophilized semi-IPN hydrogels from the CC and CCH series are presented in Figure 5 and Figure 6, respectively, while those of SHC-IPNs and of the simple polyVBTAC hydrogel are shown in Figure 7. The morphology of all IPNs illustrates a favorable porous structure for water adsorption. However, the morphology changes with the increase in VBTAC for all the three series of IPN, from a dense porous network (IPN_2_) to a compact and smooth network (IPN_4_) that more closely resembles that of poly(VBTAC) alone. Therefore, it can be noted that the three series of semi-IPN show a good intercalation of the compounds in the final structure due to their smooth appearance and defect-free areas.

### 3.4. Swelling Degree Study (SD) for the Semi-IPN Hydrogels

The influence of the VBTAC concentration and the chitosan type (CC, CCH and SHC) used for the synthesis of IPNs was also evaluated by determining the degree of swelling at equilibrium. Figure 8 shows a fast mass increase in hydrogels in the first 3 h after their immersion in liquid (water). In the case of polyVBTAC, in addition to the initial fast increase, the equilibrium swelling is reached much faster compared to the rest of the samples containing chitosan. Figure 9 shows the maximum degree of swelling (SD_MAX_) for the analyzed hydrogels. Given that semi-IPN hydrogels were synthesized using the same amount of crosslinking agent relative to the VBTAC content, it can be observed that with increasing VBTAC concentration, the SD_MAX_ values decrease considerably from IPN_2_ to IPN_3_ and to IPN_4_, particularly for the CC and SHC series, thus indicating a much more compact interpenetrating structure which tends to absorb a smaller amount of fluid. An interesting swelling behavior is observed for CCH-IPNs, whereas very little differences are observed when increasing the VBTAC concentration, which can indicate a rather limited interpenetration of networks in this case [59,60].

The type of chitosan also affected the swelling behavior of samples, in which case SHC-IPNs seem to have a more controlled and narrower range of variation, from 128 to 105 and 69, for IPN_2_, IPN_3_ and IPN_4_, respectively, versus CC-IPNs with a significant drop in SD between IPN_2_ (157) and IPN_3_ (96), followed by a moderate drop for IPN_4_ (75). Although the SD_MAX_ of 157 for the three studied series is given by CC-IPN_2_, all the samples containing SHC presented values close to those containing CC, thus supporting the hypothesis of using calcium-carbonate-enriched chitosan from shrimp carcass waste for ecological applications such as the removal of bacteria from wastewater. Another noteworthy fact refers to the fragmentation of hydrogels. For all three series, slight fragmentations were observed after 24 h, particularly for the semi-IPNs with CC and CCH, which is why the swelling experiments were performed only for the 0–1400 min time range.

### 3.5. Bacteriological Evaluation Results

Table 2 summarizes the results obtained for the bacteriological tests performed according to the method described previously. As expected, the bactericidal effect of the polyVBTAC reference was quite notable. Yet, by keeping in mind that one of the targets of the study is delivering a cost-effective bactericidal material, it can be stated that some of the semi-IPNs containing VBTAC registered very good bactericidal properties, similar to the reference (polyVBTAC). In this respect, the efficiency of semi-IPN hydrogels for Gram-negative bacteria retention was quite high for SHC-IPNs, indicating a reduction of up to 51.3% of total coliforms and 53.0% of total *C. perfringens* for SHC-IPN_4_, 18.3% of total coliforms and 27.2% of total *C. perfringens* for SHC-IPN_3_, and 13.2% of total coliforms and 21.3% of total *C. perfringens* for SHC-IPN_2_. Meanwhile, hydrogels containing CC and CCH, with the series representatives CC-IPN_3_ and CCH-IPN_3_, showed a decrease of only 14.6% and 8.5%, respectively, for total coliforms, and 12.5% and 6.3%, respectively, for total *C. perfringens*. Hence, CCH-based semi-IPNs registered the lowest bacterial retention rate of all trial sets, but were in agreement with the values recorded for the SD. Therefore, as observed for the other series as well, the SD is also a major factor of influence upon the efficiency of the semi-IPN hydrogels. In addition to the SD, the significant reduction capacity for bacteria registered for the samples in the series of semi-IPNs based on SHC may also be linked to the content of native minerals present in the chitosan structure.

### 3.6. Removal Efficiency of Waterborne Pathogens

A panel of species-specific genetic markers was applied for the quantification of the most representative waterborne microbial pathogens in raw WW samples and the ones treated with the different hydrogels developed in this study. These markers the target bacterial pathogens of both Gram-positive (*E. faecalis* and *C. perfringens*) and Gram-negative (S. Typhimurium, *C. jejuni*, *L. pneumophila*, STEC, and *Shigella* spp.) bacteria as well as protozoan pathogens (*C. parvum* and *G. lamblia*). Since *C. jejuni*, *C. parvum* and *G. lamblia* were not detected in raw WW samples, they were excluded from downstream examination. The pathogen removal rates of each trial set were estimated from the molecular quantification data and are presented in Table 3a,b. The polyVBTAC, chosen as a reference, displayed the anticipated bactericidal effects on all detected pathogens, representing a proper benchmark for overall evaluation. In comparison, the material preparations (semi-IPN hydrogels) from each trial series exhibited antibacterial efficacy to varying degrees, among which some even exerted a superior performance to polyVBTAC. It is noteworthy that the series of SHC-IPNs, especially SHC-IPN_2_ and SHC-IPN_3_, demonstrated distinctly higher effectiveness than polyVBTAC for the elimination of *E. faecalis* and *S.* Typhimurium by 100% (Table 3a). Moreover, both formulations indicated a reduction in *C. perfringens* by 78% and 63%, respectively, which were higher than polyVBTAC at 51%. In addition, SHC-IPN_2_ showed a similar strength antagonizing *L. pnuemophila* and *Shigella* spp. to polyVBTAC. With regards to curbing STEC, SHC-IPN_2_ and, particularly, SHC-IPN_3_ displayed comparable efficacy to polyVBTAC, with a considerable reduction of *stx1*, *stx2* and *eae*, ranging from 35–53% (Table 3b). As revealed in our molecular assay, CCH-derived semi-IPNs represent the second-best performers against pathogens. Notably, CCH-IPN_2_ and CCH-IPN_3_ manifested a higher removal efficacy than polyVBTAC on *E. faecalis*, *C. perfringens* and *Shigella* spp. (Table 3a,b). *S.* Typhimurium and *L. pneumophila* were better removed by CCH-IPN_2_ and CCH-IPN_4_, respectively, in comparison to polyVBTAC. In addition, all CCH-derived semi-IPNs revealed the highest removal rate of STEC *stx2*. In regard to the CC-IPNs series, CC-IPN_2_ and CC-IPN_4_ illustrated closer and/or similar antibacterial activity to polyVBTAC. In contrast, CC-IPN_3_ displayed the lowest effectiveness among all examined materials. Overall, based on the molecular characterization, hydrogels based on the chitosan prepared from shrimp shell, prominently SHC-IPN_2_ and SHC-IPN_3_, represent the most competitive and promising candidates for WW pathogen treatment among all tested materials. The remarkable antibacterial properties of SHC-IPN_2_ and SHC-IPN_3_ correspond well with their high SD_MAX_ indices, as measured in the swelling degree study, which provides supportive structural evidence/elucidation and indicates their great potential for use in WW treatment. Moreover, molecular assessment also unveiled that the combination of shrimp-shell chitosan and VBTAC could achieve a boosted/enhanced bactericidal effect, even when VBTAC was applied in a low amount, for instance at 0.3 g (SHC-IPN_2_) and 0.5 g (SHC-IPN_3_), being stronger than the reference polyVBTAC alone at the highest dosage (i.e., 1.0 g). This implies that the antibacterial synergy detected in the SHC-INPs series relies on both the bioactive chitosan produced from shrimp shell and the proven-bactericidal VBTAC.

## 4. Conclusions

The present study describes the successful synthesis of semi-IPN hydrogels based on commercial chitosan and experimental chitosan (via commercial chitin and shrimp shells), and quaternary ammonium polymers. The synthesized materials were physico-chemically characterized by modern techniques such as FTIR spectroscopy, thermogravimetry TGA/DTG, microscopy and the molecular screening of pathogens.

FTIR spectroscopic analysis highlighted the characteristic bands of the constituents in the semi-IPN hydrogels, while the thermal analysis pointed out the influence of VBTAC concentration upon the semi-IPNs’ thermostability. The study of the swelling degree was useful for optimizing the bacteriological methodologies but also to explain the bactericidal behavior of the semi-IPN hydrogels prepared with different types of chitosan. Although the maximum degree of swelling was registered for the hydrogels with commercial chitosan (CC), the bactericidal effect was higher for the samples containing shrimp-shell-derived chitosan (SHC), as revealed by the bacteriological and molecular assays. This provided empirical evidence that the bactericidal properties of semi-IPNs are also greatly influenced by the type of chitosan, and implicitly by the synthesis method, as this study attempts to demonstrate.

Bacteriological assays reflected the bactericidal potential of the synthesized materials, particularly for the SHC-based hydrogels, to destroy both coliforms and *C. perfringens*, by 51.3% and 53%, respectively. Advanced molecular diagnostics confirmed the findings of the bacteriological examinations. Moreover, other WW-harbored pathogens detected using species-specific markers, such as *E. faecalis*, *S.* Typhimurium, *L. pneumophila*, *Shigella* spp. and STEC, could also be effectively eliminated by SHC-IPNs, which in repeatable cases exhibited higher activity than polyVBTAC. Notably, a synergistic antibacterial effect was observed by integrating semi-IPNs with VBTAC. Both bacterial and molecular examinations and assessments pointed out that hydrogels based on the chitosan prepared from shrimp-shell wastes are feasible, cost-effective, and more importantly, possess an ideal bactericidal effectiveness pivotal for pathogen eliminations. Thus, the results of the present study support the hypothesis of using shrimp-shell waste for ecological applications and environmental engineering, namely in wastewater treatment.

## Figures and Tables

**Figure 1 polymers-15-01091-f001:**
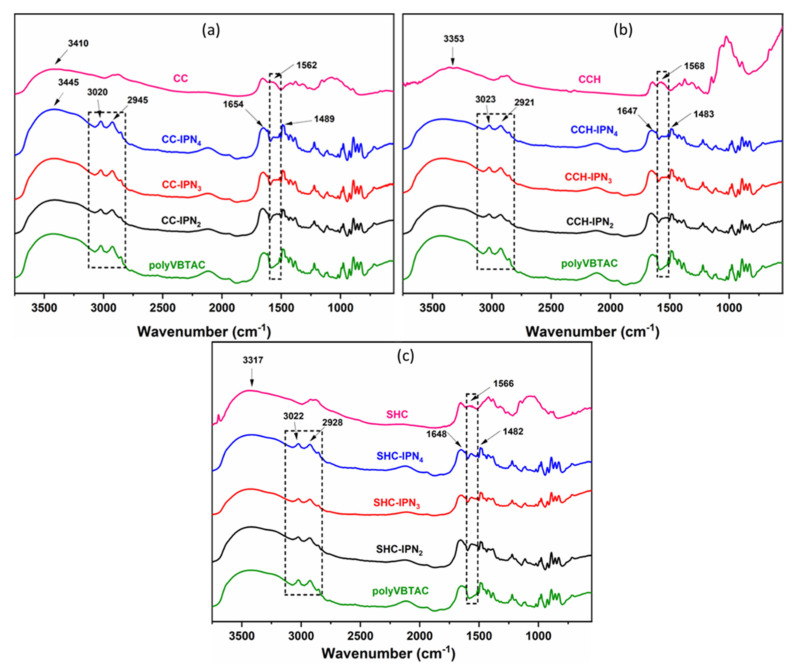
FTIR spectra of CC-IPN hydrogels (**a**), CCH-IPN hydrogels (**b**) and SHC-IPN hydrogels (**c**) compared to the FTIR spectra of polyVBTAC, CC, CCH and SHC, respectively.

**Figure 2 polymers-15-01091-f002:**
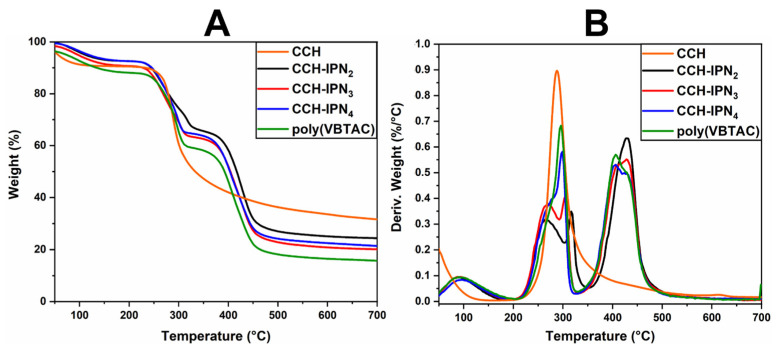
TGA (**A**) and DTG (**B**) thermograms for semi-IPN CCH-based hydrogels.

**Figure 3 polymers-15-01091-f003:**
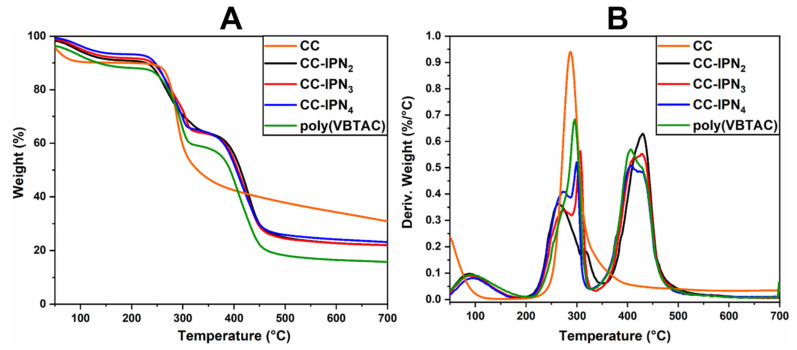
TGA (**A**) and DTG (**B**) thermograms for semi-IPN CC-based hydrogels.

**Figure 4 polymers-15-01091-f004:**
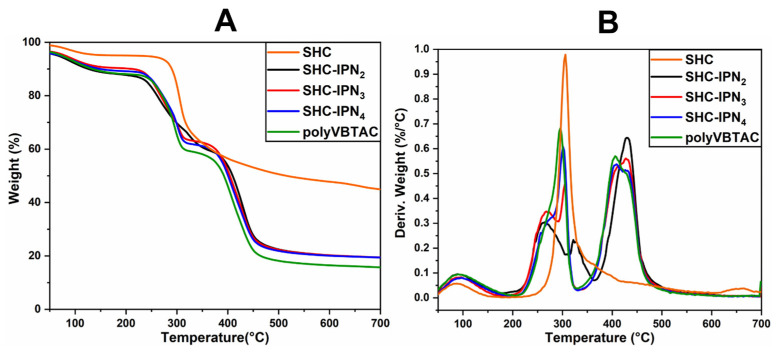
TGA (**A**) and DTG (**B**) thermograms for semi-IPN SHC-based hydrogels.

**Figure 5 polymers-15-01091-f005:**
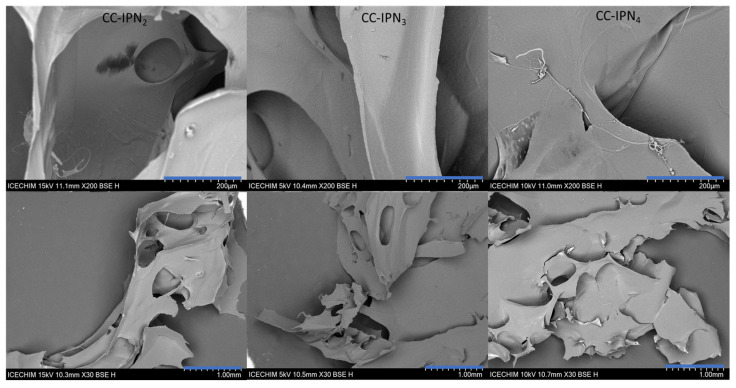
SEM micrographs of freeze-dried CC-IPNs at 200 µm (**top**) and 1 mm (**bottom**).

**Figure 6 polymers-15-01091-f006:**
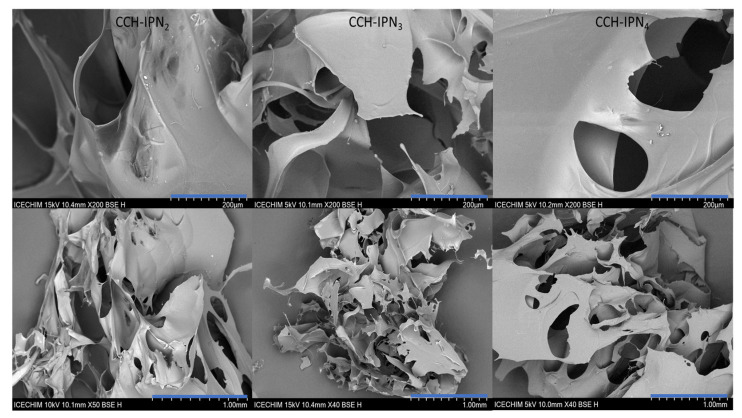
SEM micrographs of freeze-dried CCH-IPNs at 200 µm (**top**) and 1 mm (**bottom**).

**Figure 7 polymers-15-01091-f007:**
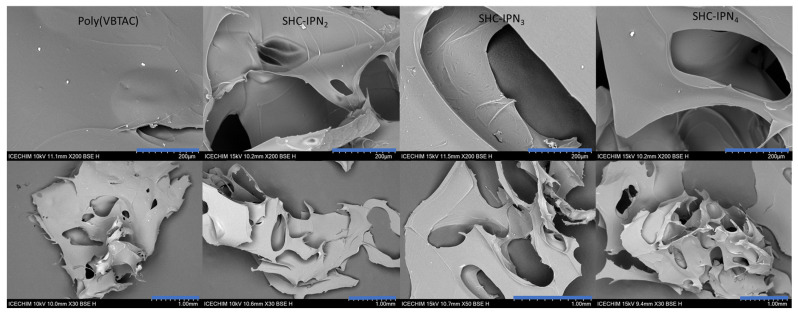
SEM micrographs of freeze-dried SHC-IPNs and polyVBTAC at 200 µm (**top**) and 1 mm (**bottom**).

**Figure 8 polymers-15-01091-f008:**
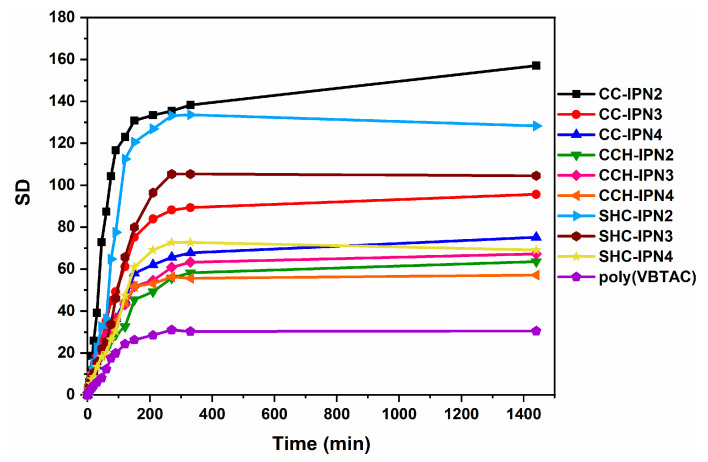
The influence of the chitosan used and VBTAC concentration upon the SD in time.

**Figure 9 polymers-15-01091-f009:**
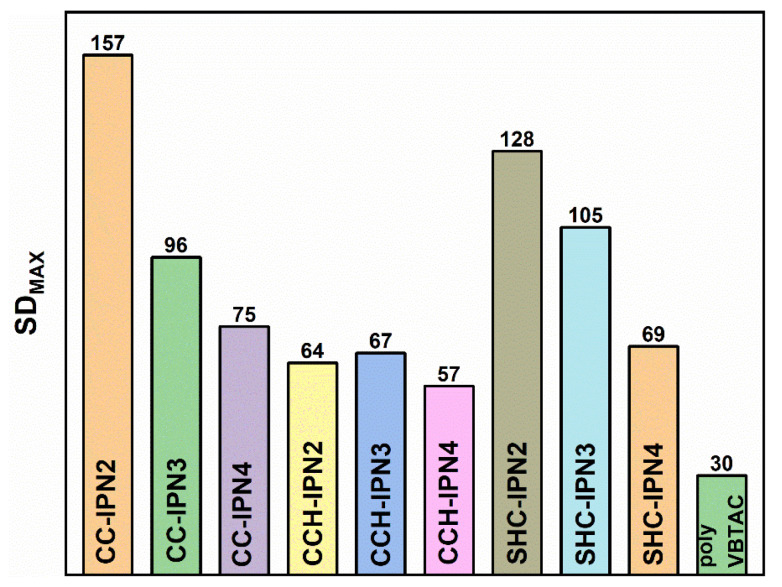
Maximum swelling degree (SD_MAX_) of the synthesized semi-IPN hydrogels.

**Table 1 polymers-15-01091-t001:** Recipes for the three series of semi-IPN hydrogels with CC, CCH and SHC.

Code	H_2_O (mL)	VBTAC (g)	MBA (wt. % rel. to Monomer Content)	ACVA (wt. % rel. to Monomer Content)	Chitosan (g)	Water: Acetic Acid (wt./wt.) **	Chitosan: VBTAC (wt./wt.)
*** CC-1**	4	0	0.05 (g)	0.01 (g)	0.01	9:1	-
**CC-IPN_2_**	4	0.3	16.66	3.33	0.01	9:1	0.1:3
**CC-IPN_3_**	4	0.5	10	2	0.01	9:1	0.1:5
**CC-IPN_4_**	4	0.7	7.14	1.42	0.01	9:1	0.1:7
*** CCH-1**	4	0	0.05 (g)	0.01 (g)	0.01	9:1	-
**CCH-IPN_2_**	4	0.3	16.66	3.33	0.01	9:1	0.1:3
**CCH-IPN_3_**	4	0.5	10	2	0.01	9:1	0.1:5
**CCH-IPN_4_**	4	0.7	7.14	1.42	0.01	9:1	0.1:7
*** SHC-1**	4	0	0.05 (g)	0.01 (g)	0.01	5:5	-
**SHC-IPN_2_**	4	0.3	16.66	3.33	0.01	5:5	0.1:3
**SHC-IPN_3_**	4	0.5	10	2	0.01	5:5	0.1:5
**SHC-IPN_4_**	4	0.7	7.14	1.42	0.01	5:5	0.1:7
**polyVBTAC**	4	1	5	1	-	-	-

* Gels contain only chitosan, 0.05 g crosslinker and 0.01 g initiator but were unstable and were excluded from the following studies; ** a higher quantity of acetic acid was needed to solubilize SHC, with higher crystallinity.

**Table 2 polymers-15-01091-t002:** Bacteriological indicators evaluated for the synthesized semi-IPN hydrogels after contact with WW and the standard error of means (SE) for each trial set.

Sample	Indicator *		Decrease Compared to Reference Sample	
	Coliforms ± SE, CFU·150 mL^−1^	*C. perfringens* ± SE, CFU·150 mL^−1^	Coliforms± SE, %	*C. perfringens* ± SE, %
**Reference WW1**	625.0	370.0		
**CC-IPN_2_**	601.0 ± 2.3	351.3 ± 2.4	3.8 ± 3.7E-01	5.1 ± 6.5E-01
**CC-IPN_3_**	534.0 ± 5.5	323.7 ± 3.5	14.6 ± 8.8E-01	12.5 ± 9.5E-01
**CC-IPN_4_**	604.7 ± 2.9	361.0 ± 2.1	3.3 ± 4.7E-01	2.4 ± 5.6E-01
**Reference WW2**	498.0	163.0		
**CCH-IPN_2_**	473.3 ± 3.8	161.0 ± 3.8	5.0 ± 7.7E-01	1.2 ± 2.3
**CCH-IPN_3_**	455.7 ± 2.9	152.7 ± 3.8	8.5 ± 5.9E-01	6.3 ± 2.4
**CCH-IPN_4_**	487.7 ± 1.3E+01	146.7 ± 4.8	2.1 ± 2.5	10.0 ± 2.9
**Reference WW3**	520.0	314.0		
**SHC-IPN_2_**	451.3 ± 9.4	247.0 ± 2.3	13.2 ± 1.8	21.3 ± 7.4E-01
**SHC-IPN_3_**	425.0 ± 7.1	228.7 ± 4.5	18.3 ± 1.4	27.2 ± 1.4
**SHC-IPN_4_**	253.3 ± 7.2	147.7 ± 4.1	51.3 ± 1.4	53.0 ± 1.3
**polyVBTAC**	208.0 ± 10.2	131.0 ± 2.9	60.0 ± 1.4	58.3 ± 1.0

* The measurements (M) were performed in triplicate and the standard error of means, SE (±), was calculated with the equation SD/(n^1/2^), where n is the number of experiments (3), SD is the standard deviation SD = (Σ(ΔM_n_^2^)/(n − 1))^1/2^, ΔM_n_ is the deviation of data from the mean value, and CFU is the colony-forming units.

**Table 3 polymers-15-01091-t003:** (**a**) Genetic-marker-based detection of pathogens as quantified in copy numbers (CN) per 100 mL with standard error of means (SE) and their removal rate (RR) in percentage for each treatment trial. (**b**) Genetic marker-based detection of pathogens as quantified in copy numbers (CN) per 100 mL with standard error of means (SE) and their removal rate (RR) in percentage for each treatment trial.

(a)								
Sample	*Enterococcus faecalis*		*Clostridium perfringens*		*Salmonella* Typhimurium		*Legionella pneumophila*	
	CN ± SE·100 mL^−1^	RR	CN ± SE·100 mL^−1^	RR	CN ± SE·100 mL^−1^	RR	CN ± SE·100 mL^−1^	RR
**WW1**	4.85E+05 ± 6.17E+01		3.62E+03 ± 1.36E-01		4.00E+06 ± 8.57E+02		4.00E+02 ± 1.16E-01	
**CC-IPN_2_**	9.26E+04 ± 1.47E+01	80%	2.40E+03 ± 2.81E-01	34%	9.51E+05 ± 1.30E+00	76%	2.57E+02 ± 1.60E-01	36%
**CC-IPN_3_**	3.41E+05 ± 2.61E+01	25%	3.90E+03 ± 4.31E-01	0%	4.16E+06 ± 3.83E+02	0%	4.57E+02 ± 5.97E-02	0%
**CC-IPN_4_**	5.14E+05 ± 6.44E+01	0%	1.59E+03 ± 1.89E-01	56%	5.58E+06 ± 1.62E+02	0%	6.97E+01 ± 1.29E-01	83%
**WW2**	5.91E+05 ± 8.28E+01		2.18E+03 ± 1.00E-01		3.70E+06 ± 2.67E+01		9.90E+02 ± 8.75E-02	
**CCH-IPN_2_**	5.61E+04 ± 3.50E+00	91%	1.01E+03 ± 3.87E-01	54%	4.51E+03 ± 2.57E-01	100%	6.17E+02 ± 1.45E-01	38%
**CCH-IPN_3_**	6.82E+04 ± 5.90E+00	88%	4.81E+02 ± 1.54E-01	78%	5.96E+05 ± 1.88E+01	84%	4.10E+02 ± 6.22E-02	59%
**CCH-IPN_4_**	1.68E+05 ± 2.22E+01	72%	1.43E+03 ± 2.09E-02	34%	1.87E+06 ± 1.13E+02	50%	6.64E+01 ± 1.03E-01	93%
**WW3**	5.25E+05 ± 4.25E+01		1.27E+03 ± 1.22E-01		4.86E+05 ± 2.65E+00		9.00E+02 ± 5.76E-02	
**SHC-IPN_2_**	0.00E+00 ± 0.00E+00	100%	2.85E+02 ± 1.01E-01	78%	1.59E+03 ± 1.20E-01	100%	3.17E+02 ± 2.65E-01	65%
**SHC-IPN_3_**	0.00E+00 ± 0.00E+00	100%	4.74E+02 ± 8.09E-02	63%	7.46E+02 ± 1.90E-01	100%	7.67E+02 ± 4.22E-01	15%
**SHC-IPN_4_**	0.00E+00 ± 0.00E+00	100%	1.28E+03 ± 3.43E-01	0%	2.48E+04 ± 8.36E+00	95%	4.51E+02 ± 5.38E-02	50%
**polyVBTAC**	1.35E+05 ± 1.25E+01	74%	1.08E+03 ± 3.04E-01	51%	4.23E+04 ± 1.84E+00	91%	1.56E+02 ± 5.51E-03	84%
(**b**)								
Sample	*Shigella* spp.		Shiga toxin-producing *Escherichia coli* (STEC)					
			*stx1*		*stx2*		*eae*	
	CN ± SE·100 mL^−1^	RR	CN ± SE·100 mL^−1^	RR	CN ± SE·100 mL^−1^	RR	CN ± SE·100 mL^−1^	RR
**WW1**	4.00E+02 ± 1.33E-01		4.74E+06 ± 3.20E+01		1.80E+05 ± 1.82E+00		1.51E+04 ± 1.55E+00	
**CC-IPN_2_**	2.57E+02 ± 7.27E-02	36%	2.57E+06 ± 1.22E+01	46%	1.15E+05 ± 4.95E+00	36%	1.13E+04 ± 9.67E-01	25%
**CC-IPN_3_**	4.57E+02 ± 2.65E-01	0%	2.68E+06 ± 1.01E+02	44%	1.21E+05 ± 5.29E+00	33%	1.07E+04 ± 2.20E-01	29%
**CC-IPN_4_**	6.97E+01 ± 1.80E-01	83%	2.92E+06 ± 4.00E+01	38%	1.20E+05 ± 5.91E+00	33%	1.30E+04 ± 4.40E-01	14%
**WW2**	7.86E+03 ± 9.47E-01		4.22E+06 ± 2.20E+01		1.79E+05 ± 1.19E+00		1.62E+04 ± 1.55E+00	
**CCH-IPN_2_**	9.50E+02 ± 5.91E-01	88%	2.29E+06 ± 3.42E+02	14%	6.69E+04 ± 1.99E+00	63%	9.98E+03 ± 7.45E-01	39%
**CCH-IPN_3_**	1.27E+03 ± 4.76E-01	84%	2.18E+06 ± 8.07E+01	18%	7.08E+04 ± 3.57E-01	60%	1.37E+04 ± 1.05E-01	15%
**CCH-IPN_4_**	2.35E+03 ± 3.30E-01	70%	2.71E+06 ± 1.93E+02	0%	8.85E+04 ± 6.73E+00	50%	9.89E+03 ± 1.07E+00	39%
**WW3**	1.99E+03 ± 6.81E-02		3.81E+06 ± 2.80E+01		1.80E+05 ± 1.10E+00		1.59E+04 ± 3.49E-01	
**SHC-IPN_2_**	7.03E+02 ± 2.52E-01	65%	2.49E+06 ± 1.14E+01	35%	9.10E+04 ± 7.75E+00	49%	3.09E+04 ± 6.24E-01	0%
**SHC-IPN_3_**	1.02E+03 ± 5,66E-02	49%	2.24E+06 ± 1,11E+02	41%	8.53E+04 ± 1.23E+01	53%	7.71E+03 ± 1.06E+00	52%
**SHC-IPN_4_**	1.31E+03 ± 5.09E-03	34%	2.53E+06 ± 2.64E+02	34%	1.08E+05 ± 6.52E+00	40%	9.99E+03 ± 1.49E+00	37%
**polyVBTAC**	2.01E+03 ± 4.72E-01	74%	2.31E+06 ± 3.60E+01	46%	9.25E+04 ± 2.77E+00	49%	6.86E+03 ± 1.87E+00	55%

## Data Availability

The data presented in this study are available on request from the corresponding authors.

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
