# Peer review of "Introducing Semi-Interpenetrating Networks of Chitosan and Ammonium-Quaternary Polymers for the Effective Removal of Waterborne Pathogens from Wastewaters"

_polymers, 2023, doi:10.3390/polym15051091_

Round 1
Reviewer 1 Report
Neblea et al. presented a method to synthesize semi-IPN hydrogels based on commercial and experimental obtained chitosan and quaternary ammonium polymer (pVBTAC) for wastewater treatment application. The properties of the hydrogels were thoroughly characterized and elucidated. However, there are still some suggestions for this manuscript before it’s ready for publication,
1. In 3.1. FTIR Spectroscopy of semi-IPN hydrogels, besides 1489 cm-1 vibration peak, the quaternary ammonium groups may also show up at ~ 960 cm-1, investigation of this peak should be included in the manuscript. Also, the peak intensity at 1489 cm-1 is not a function of VBTAC concentration in the hydrogel, the CCH-IPN hydrogels showed undesired similar intensity at this peak, the authors should normalize the FTIR spectra or rescan the spectra.
2. In 3.4. Swelling degree study (SD) for the semi-IPN hydrogels, an analysis based on the volumetric swelling ratio should be presented. Since this hydrogel was designed for wastewater treatment, the volumetric swelling behavior is crucial to its performance over long-term service. The authors may consider prepare dried hydrogel specimen with well-defined geometry then compare with its hydrated state.
3. In 3.5. Bacteriological evaluation results section, due to the fouling issue of cationic polymers, the killed bacteria can attach to the hydrogel surface and form a barrier between pVBTAC and incoming viable bacteria and reduce the killing property of the hydrogel. How much of the antibacterial property will be retained after multiple cycles of service? Will the dead bacterial debris from previous duty cycle compromise the efficiency for the next cycles? Please provide more characterizations.
Author Response
- The manuscript in red and marked up using the “Track Changes” function contains all changes added according to reviewers’ suggestions. Also, the supplementary file has been modified accordingly.
- All the references are relevant to the contents of the manuscript. Also new references were added as a consequence of additional explanations required by reviewers which led to changes in the reference’s countdown.
ANSWER TO SPECIFIC QUESTIONS OF REVIEWER #1
Comments from Reviewer #1: Neblea et al. presented a method to synthesize semi-IPN hydrogels based on commercial and experimental obtained chitosan and quaternary ammonium polymer (pVBTAC) for wastewater treatment application. The properties of the hydrogels were thoroughly characterized and elucidated. However, there are still some suggestions for this manuscript before it’s ready for publication.
- In 3.1. FTIR Spectroscopy of semi-IPN hydrogels, besides 1489 cm-1 vibration peak, the quaternary ammonium groups may also show up at ~ 960 cm-1, investigation of this peak should be included in the manuscript. Also, the peak intensity at 1489 cm-1 is not a function of VBTAC concentration in the hydrogel, the CCH-IPN hydrogels showed undesired similar intensity at this peak, the authors should normalize the FTIR spectra or rescan the spectra.
The authors thank the reviewer for the observation. In this regard, the presence of the peak at 960 cm-1 which indicates the presence of quaternary ammonium groups was also noted in the manuscript (section 3.1. FTIR Spectroscopy of semi-IPN hydrogels) and a new reference was added to support the affirmation [Yudovin-Farber, I., Beyth, N., Weiss, E.I. et al. Antibacterial effect of composite resins containing quaternary ammonium polyethyleneimine nanoparticles. J Nanopart Res 12, 591–603 (2010). https://doi.org/10.1007/s11051-009-9628-8] as follows: “In addition to the bands specific to chitosan, the samples show bands at 1489 cm-1 and at 960 cm-1 indicating the quaternary ammonium group vibration, C-N+(CH3)3, specific for the synthetic polymer VBTAC and a peak at 3020 cm-1 specific to C-H vibration from the macromolecular chain of poly(VBTAC) [58] (Figure 1a, b and c).”
The rescanned FTIR spectra showed the similar intensity at the indicated peak (1489 cm-1) and therefore we didn’t modify the specified Figures. However, this should not affect in any way the value of the measurement, giving the fact that we considered FTIR for qualitative measurements rather than quantitative.
- In 3.4. Swelling degree study (SD) for the semi-IPN hydrogels, an analysis based on the volumetric swelling ratio should be presented. Since this hydrogel was designed for wastewater treatment, the volumetric swelling behavior is crucial to its performance over long-term service. The authors may consider prepare dried hydrogel specimen with well-defined geometry then compare with its hydrated state.
Indeed, for this kind of application of hydrogels, the volumetric swelling behavior is determinant to its performance over time. The authors appreciate the reviewer’s suggestion and will consider this parallel investigation for the future study. But for the time being, given the fact that these samples are in a preliminary stage, we believe that the swelling degree (SD) by weight represents the most convenient way to evaluate their behavior in water.
- In 3.5. Bacteriological evaluation results section, due to the fouling issue of cationic polymers, the killed bacteria can attach to the hydrogel surface and form a barrier between pVBTAC and incoming viable bacteria and reduce the killing property of the hydrogel. How much of the antibacterial property will be retained after multiple cycles of service? Will the dead bacterial debris from previous duty cycle compromise the efficiency for the next cycles? Please provide more characterizations.
The authors would like to mention that the hydrogel samples have been considered for single-use service. However, the project is ongoing and our research will also target methods for reconditioning and reuse of hydrogels for at least 2 cycles, but also biodegradability studies due their high content of chitosan. Therefore, these are questions to be answered in the future studies. About the bacteria attachment, we believe that most of the killing occurs after there are being adsorbed in the hydrogel. We did not mention anything about this mechanism because we need to have some validations, at least from CryoTEM analysis. Since such analysis require more time, we decided to include the new findings in a future publication.
Reviewer 2 Report
Finding more efficient and cost-effective materials for the removal of waterborne pathogens and bacteria, from the wastewaters, is a stringent need nowadays. The authors of this manuscript used as a strategy for this purpose the synthesis of semi-IPN hydrogels having cross-linked poly(VBTAC) as the network and three types of chitosan as entrapped polycation. However, the manuscript must be carefully improved because of too many mistakes and ambiguous formulations. Some examples of the changes/corrections required in this manuscript are mentioned below.
· Please, use in the whole manuscript “semi-interpenetrating networks” (used first in the title) and not “semi-interpenetrated networks” as in the Abstract; it is not correct “semi-interpenetrated hydrogel” placed as keyword.
· Abstract: It is not clear what does it mean “mineral enriched chitosan”! Lines 26-27, “examined and characterized through systematic physio-chemical methods” must be rewritten! Line 30, what is the meaning of “WW”?
· Introduction: line 58, remove brackets in “(semi)-interpenetrating polymer networks”! Line 60, remove brackets in “(semi)-IPN! Line 69, “from semi-IPNs”, please correct to make sense! Line 85, what does it mean “56-D-glucose”? Lines 86-87: “by dissolving in a non-solvent”, it is not clear! Lines 94-110, please rewrite and compress this part removing repetitions! Actually, the bactericidal properties of quaternary ammonium salts have been well described long time ago! The authors must present, from the beginning, in a clearer manner, the process for the preparation of “mineral enriched chitosan” because the reader cannot find any information about it in the experimental part!
· Materials and Methods: the following text “chitosan synthesized in previous works from commercial chitin by deacetylation process (CCH, 77% deacetylation degree, Mn=4.7291·105 g/mol [47]) and chitosan extracted in previous works from shrimp shells (SHC, 76% deacetylation degree, Mn=9.058·103 g/mol” is presented in lines 133-136; there are no information about the mineral enrichment of chitosan. Furthermore, the values of Mn of chitosan are not clear.
· The content of cross-linker and initiator must be related with the monomers and not with water content, as it is written in line 147 and in Table 1!
· In the whole manuscript the abbreviation of the composite hydrogels as CC-IPN, CCH-IPN and SCH-IPN is ambiguous because the reader can see only “IPN” and not “semi-IPN” as these hydrogels have been designed. Moreover, the samples CC-IPN1, CCH-IPN1 and SHC-IPN1 in Table 1 have been prepared only with chitosan, without any cross-linker? The code of these samples is also ambiguous! There is no IPN.
· Line 188, the abbreviation of “swelling degree” is missing; Eq. (1) does not give the values of SD, which is just a ratio, but the water content of hydrogel!
· The authors should provide SEM micrographs with a lower magnification as the information about the hydrogel morphology to be more reliable.
Replace “,” with “.” in Tables 1, 3a and 3b!
Author Response
- The manuscript in red and marked up using the “Track Changes” function contains all changes added according to reviewers’ suggestions. Also, the supplementary file has been modified accordingly.
- All the references are relevant to the contents of the manuscript. Also new references were added as a consequence of additional explanations required by reviewers which led to changes in the reference’s countdown.
ANSWER TO SPECIFIC QUESTIONS OF REVIEWER #2
Comments from Reviewer #2: Finding more efficient and cost-effective materials for the removal of waterborne pathogens and bacteria, from the wastewaters, is a stringent need nowadays. The authors of this manuscript used as a strategy for this purpose the synthesis of semi-IPN hydrogels having cross-linked poly(VBTAC) as the network and three types of chitosan as entrapped polycation. However, the manuscript must be carefully improved because of too many mistakes and ambiguous formulations. Some examples of the changes/corrections required in this manuscript are mentioned below.
- Please, use in the whole manuscript “semi-interpenetrating networks” (used first in the title) and not “semi-interpenetrated networks” as in the Abstract; it is not correct “semi-interpenetrated hydrogel” placed as keyword.
The authors would like to thank the reviewer for the observation. The formulation "semi-interpenetrating networks" (used first in the title) instead of "semi-interpenetrated networks" was used both in the Abstract and as a keyword.
Abstract:
- It is not clear what does it mean “mineral enriched chitosan”!
As described in the introduction section (line 118), “mineral enriched-chitosan” refers to the type of chitosan extracted from the shrimp shells without passing the demineralization stage, as described in ref. [52] (Miron, A.; Sarbu, A.; Zaharia, A.; Sandu, T.; Iovu, H.; Fierascu, R.C.; Neagu, A.-L.; Chiriac, A.-L.; Iordache, T.-V. A Top-Down Procedure for Synthesizing Calcium Carbonate-Enriched Chitosan from Shrimp Shell Wastes. Gels 2022, 8, 742, doi:10.3390/gels8110742). We tried to underline the difference between the commercial variants of chitosan and/or chitin which are previously demineralized.
- Lines 26-27, “examined and characterized through systematic physio-chemical methods” must be rewritten!
The phrase “examined and characterized through systematic physio-chemical methods” was rewritten as follows: “The new semi-IPNs were characterized for composition, thermal stability and morphology using well-known methods.”
- Line 30, what is the meaning of “WW”?
The authors refer by "WW" formulation to the wastewater samples as mentioned first in the Abstract section.
Introduction:
- Line 58, remove brackets in “(semi)-interpenetrating polymer networks”!
The formulation was modified according to the reviewer’s suggestion.
- Line 60, remove brackets in “(semi)-IPN!
The brackets have been properly removed.
- Line 69, “from semi-IPNs”, please correct to make sense!
The indicated sentence was modified for a better clarification as follows: “… to form double crosslinked interpenetrating networks of polymers (IPNs), or by polymerization/ crosslinking a monomer or a pre-polymer around an existing polymer chain [26, 27] to form semi-IPNs”.
- Line 85, what does it mean “56-D-glucose”?
The literature describes chitosan as a semicrystalline polysaccharide that contains copolymers of D-glucosamine (deacetylated units) and N-acetyl-D-glucosamine (acetylated units) linked by β-(1,4) glycosidic bonds [38-Sereni, N.; Enache, A.; Sudre, G.; Montembault, A.; Rochas, C.; Durand, P.; Perrard, M.-H.; Bozga, G.; Puaux, J.-P.; Delair, T.; et al. Dynamic Structuration of Physical Chitosan Hydrogels. Langmuir 2017, 33, 12697–12707, doi:10.1021/acs.langmuir.7b02997]. Therefore, for any other misunderstandings the specified phrase was modified as follows: “Moreover, chitosan is soluble in acidic medium in the presence of amine groups from the 56 D-glucosamine units [38]”.
- Lines 86-87: “by dissolving in a non-solvent”, it is not clear!
The phrase was reformulated as follows: “It can be crosslinked by increasing the pH of the solution or by casting the solution in a non-solvent, ideal for wastewater treatment and heavy metals retention, as described by Bhullar et al. [39] in their paper regarding semi-IPNs of chitosan/acrylic acid and thiourea hydrogels.”
- Lines 94-110, please rewrite and compress this part removing repetitions! Actually, the bactericidal properties of quaternary ammonium salts have been well described long time ago! The authors must present, from the beginning, in a clearer manner, the process for the preparation of “mineral enriched chitosan” because the reader cannot find any information about it in the experimental part!
The paragraph “Compounds with quaternized amines such as pyridinium, triethylammonium bromide or vinyl benzyl trimethylammonium chloride are well-known to possess antimicrobial activity against specific microorganisms such as Staphylococcus aureus, Escherichia coli, Candida albicans, Vibrio fischeri and Staphylococcus albus [45].” was rephrased and compressed to remove any repetions.
The author would like to thank the reviewer for the precious comments and would like to mention that the process for obtaining the mineral enriched-chitosan is not presented in the experimental part because this material has been already published in a previous work as indicated by reference [52] (Miron, A.; Sarbu, A.; Zaharia, A.; Sandu, T.; Iovu, H.; Fierascu, R.C.; Neagu, A.-L.; Chiriac, A.-L.; Iordache, T.-V. A Top-Down Procedure for Synthesizing Calcium Carbonate-Enriched Chitosan from Shrimp Shell Wastes. Gels 2022, 8, 742, doi:10.3390/gels8110742). However, a short description about the process was given in Section 2.2. so that readers have a better understanding about the present study:
“CCH and SHC were prepared previously starting from either commercial chitin of shrimp shells, respectively, and described by Miron et al. [51]. In brief, chitosan synthesis starting from shrimp shells involved only deproteinization and deacetylation, compared to the classical procedure which also included a demineralization process. Shrimp shells were first cleaned with water, desiccated with NaCl in direct sun light, and lyophilized. Deproteinization of dried shrimp shells was performed with an aqueous solution of 1 M NaOH (SH: NaOH = 1:15, w/v) at room temperature, under magnetic stirring (200 rpm) for 24 h, in order to obtain a mineral-enriched chitin. In the following step, deacetylation of chitin was performed using a NaOH 50% solution (chi-tin: NaOH = 1:20, w/v), resulting in a mineral-enriched chitosan (SHC). The reaction was conducted in one batch for 6 h, at 80 °C, under magnetic stirring (400 rpm). After each procedure, the samples were washed with distilled water until neutral pH, filtered and lyophilized. CCH was synthesized using the same deacetylation protocol described previously for SHC, but using commercial chitin instead.”
Materials and Methods:
- The following text “chitosan synthesized in previous works from commercial chitin by deacetylation process (CCH, 77% deacetylation degree, Mn=4.7291·105 g/mol [47]) and chitosan extracted in previous works from shrimp shells (SHC, 76% deacetylation degree, Mn=9.058·103 g/mol” is presented in lines 133-136; there are no information about the mineral enrichment of chitosan. Furthermore, the values of Mn of chitosan are not clear.
Thank you very much for the observations. We modified the indicated errors in the manuscript and corrected the Mn of chitosan. Also, the procedures for preparing the mineral enriched chitosan was given in brief in section 2.2.
- The content of cross-linker and initiator must be related with the monomers and not with water content, as it is written in line 147 and in Table 1!
The crosslinker and initiator content were calculated related to the monomer and this was specified in section “2.1. Materials for the synthesis of semi-IPN hydrogels”. The values were modified in Table 1, as well.
- In the whole manuscript the abbreviation of the composite hydrogels as CC-IPN, CCH-IPN and SCH-IPN is ambiguous because the reader can see only “IPN” and not “semi-IPN” as these hydrogels have been designed. Moreover, the samples CC-IPN1, CCH-IPN1 and SHC-IPN1 in Table 1 have been prepared only with chitosan, without any cross-linker? The code of these samples is also ambiguous! There is no IPN.
The authors would like to thank the reviewer for the suggestion. But we should mention that all the samples are semi-IPN, as mentioned in the whole manuscript whenever we did not use the exact codes of the samples (their design is clearly described in Section 2.2., as well). Therefore, we believe that no confusions will be made, as the sample codes were only chosen in order to simplify the notations in the manuscript, especially in the figures and tables. As for the samples CC-IPN1, CCH-IPN1 and SHC-IPN1, there are indeed without crosslinker and VBTAC, and their notations were modified accordingly in Table 1, section “2.2. Synthesis and purification of semi-IPN hydrogels”.
- Line 188, the abbreviation of “swelling degree” is missing; Eq. (1) does not give the values of SD, which is just a ratio, but the water content of hydrogel!
The swelling degree was not given in %, as usually expressed, because of the very high values which would have been difficult to follow in the main text. Therefore, we preferred to use Eq. 1, as used by other authors as well [Vieira R.M., Vilela P.B., Becegato V.A., Paulino A.T., Chitosan-based hydrogel and chitosan/acid-activated montmorillonite composite hydrogel for the adsorption and removal of Pb+2 and Ni+2 ions accommodated in aqueous solutions, J Environ Chem Eng, 2018, 6, 2713-2723, https://doi.org/10.1016/j.jece.2018.04.018; Damiri F., Bachra Y., Bounacir C., Laaraibi A., Berrada M., Synthesis and Characterization of Lyophilized Chitosan-Based Hydrogels Cross-Linked with Benzaldehyde for Controlled Drug Release, Journal of Chemistry, 2020, ID8747639, https://doi.org/10.1155/2020/8747639]
- The authors should provide SEM micrographs with a lower magnification as the information about the hydrogel morphology to be more reliable.
The authors thank the reviewer for the suggestions. In this regard, the SEM micrographs at 500 µm scale were replaced in the Figures 5-7 with figures of lower magnification (1 mm scale).
- Replace “,” with “.” in Tables 1, 3a and 3b!
Tables 1, 3a and 3b were modified accordingly.
Reviewer 3 Report
The article reported a series of semi-interpenetrated polymer network hydrogels composed of chitosan and ammonium-quaternary polymers to remove waterborne pathogens in wastewater, which can attract attentions in the field of water disinfection. However, some problems are necessary to be solved to improve this manuscript before it can be considered for publication. Detailed comments are listed below.
1. This manuscript declared that absorption process is an often-applied method for water treatment. However, the "absorption" is different from the "adsorption", and the "adsorption" should be correct in common wastewater applications. Moreover, chitosan is also an adsorbent for pollutants removal, rather than an absorbent material. This conceptual noun should be revised in the whole manuscript, especially in the introduction section. The relevant references discussing this concept are suggested to be read and may be added. Matter 2019, 1, 115-155 and Materials Horizons, 2022,9, 2496-2517, and Journal of Cleaner Production 2021, 291 125880.
2. In page 4, line 159, what does the "n=1÷4" mean? Maybe it should be "n=1-4". Please clarify this point.
3. In table 1, the subscript of H2O was not labeled. Please check.
4. In the section of 2.2. Synthesis and purification of semi-IPN hydrogels, the hydrogel samples without VBTAC were named as CC-IPN1, CCH-IPN1 and SHC-IPN1. However, in figure 1-4 and table S1-S2, all these samples without VBTAC were expressed as CC,CCH, and SHC? Please check these names of samples.
5. What does the "PEGDA" in the caption of table S1 mean?
6. The scale bars of SEM images in figure 5-7 are suggested to be labeled more clearly. For example, the scale bars in following published work can be referenced. Chemical Engineering Journal 2022, 427 130905.
7. In page 12, line 374, the CCH-IPN3 sample was described to show a decrease of 2.1% and 10% for total coliforms and C. perfringens. However, according to the values in table 2, it seemed that the sample exhibiting this performance should be CCH-IPN4. Please check.
8. Some phenomenon in table 2 is necessary to be discussed more. For example, the polyVBTAC sample demonstrated an excellent bactericidal effect. However, after introducing the VBTAC into the chitosan hydrogel system, the samples did not exhibit increasing bactericidal effect with the increasing content of VBTAC in hydrogels. Please explain why?
9. More discussion could be provided. For example, why several types of chitosan were selected to conducted this study? And why the SHC-IPN sample exhibited better bactericidal effectiveness?
10. In the introduction section, several water treatment methods have been mentioned. Besides, solar steam generation is also an emerging method used for clean water production in recent years. The recent published works may be added, such as Adv. Mater. 2023, 35, 2209015, https://doi.org/10.1016/j.scib.2023.01.017
11. The hydrogels prepared in this work showed effective bactericidal effect for wastewater treatment. Recently, it was also reported that such antibacterial property of polymeric materials could facilitate the stable and durable clean water remediation performance (Materials Horizons, 2023,10, 268-276). Therefore, the bactericidal benefit of this hydrogel consisting of chitosan and ammonium-quaternary hydrogels may be also mentioned in the discussion.
Author Response
- The manuscript in red and marked up using the “Track Changes” function contains all changes added according to reviewers’ suggestions. Also, the supplementary file has been modified accordingly.
- All the references are relevant to the contents of the manuscript. Also new references were added as a consequence of additional explanations required by reviewers which led to changes in the reference’s countdown.
ANSWER TO SPECIFIC QUESTIONS OF REVIEWER #3
Comments from Reviewer #3: The article reported a series of semi-interpenetrated polymer network hydrogels composed of chitosan and ammonium-quaternary polymers to remove waterborne pathogens in wastewater, which can attract attentions in the field of water disinfection. However, some problems are necessary to be solved to improve this manuscript before it can be considered for publication. Detailed comments are listed below.
- This manuscript declared that absorption process is an often-applied method for water treatment. However, the "absorption" is different from the "adsorption", and the "adsorption" should be correct in common wastewater applications. Moreover, chitosan is also an adsorbent for pollutants removal, rather than an absorbent material. This conceptual noun should be revised in the whole manuscript, especially in the introduction section. The relevant references discussing this concept are suggested to be read and may be added. Matter 2019, 1, 115-155 and Materials Horizons, 2022,9, 2496-2517, and Journal of Cleaner Production2021, 291 125880.
The authors would like to thank the reviewer for the observations and mention they modified the manuscript accordingly and added the relevant references suggested.
- In page 4, line 159, what does the "n=1÷4" mean? Maybe it should be "n=1-4". Please clarify this point.
“n=1÷4” was used to indicate the notation used for the samples, “n” representing the values from 1 to 4.
- In table 1, the subscript of H2O was not labelled. Please check.
The subscript of H2O was modified according to the reviewer's suggestion.
- In the section of 2.2. Synthesis and purification of semi-IPN hydrogels, the hydrogel samples without VBTAC were named as CC-IPN1, CCH-IPN1 and SHC-IPN1. However, in figure 1-4 and table S1-S2, all these samples without VBTAC were expressed as CC,CCH, and SHC? Please check these names of samples.
The authors would like to thank the reviewer for the observations and mention that the CC, CCH and SHC notations are for the chitosan samples, not for the hydrogel samples. The chitosan samples were analysed separately in order to show the presence of the characteristic peaks and the specific degradation temperatures. Therefore, the captions were corrected as follows: “Table S1. The main characteristic bands of semi-IPN hydrogels based on monomer VBTAC, crosslinker MBA and chitosan and of the three types of chitosan (CC, CCH and SHC)” and “Table S2. Specific decomposition temperatures and mass loss of the chitosan-based hydrogels and of the three types of chitosan (CC, CCH and SHC)”. Due to multiple revisions by the authors, another error to the title was found in the Supplemental file as well and corrected accordingly: “Introducing semi-interpenetrating networks of chitosan and ammonium-quaternary polymers for the effective removal of waterborne pathogens from wastewaters”
- What does the "PEGDA" in the caption of table S1 mean?
This notation was an error and the authors corrected the caption accordingly “The main characteristic bands of semi-IPN hydrogels based on monomer VBTAC, crosslinker MBA and chitosan and of the three types of chitosan (CC, CCH and SHC).”
- The scale bars of SEM images in figure 5-7 are suggested to be labeled more clearly. For example, the scale bars in following published work can be referenced. Chemical Engineering Journal 2022, 427 130905.
The scale bars of SEM micrographs From Figures 5-7 were labelled more clearly by adding a thick blue bar on the initial pointed bar.
- In page 12, line 374, the CCH-IPN3 sample was described to show a decrease of 2.1% and 10% for total coliforms and C. perfringens. However, according to the values in table 2, it seemed that the sample exhibiting this performance should be CCH-IPN4. Please check.
Thank you for the observation. The error was corrected “…CC-IPN3 and CCH-IPN3, showed a decrease of only 14.6% and 8.5%, respectively, for total coliforms, and 12.5% and 6.3%, respectively, for total C. perfringens.”
- Some phenomenon in table 2 is necessary to be discussed more. For example, the polyVBTAC sample demonstrated an excellent bactericidal effect. However, after introducing the VBTAC into the chitosan hydrogel system, the samples did not exhibit increasing bactericidal effect with the increasing content of VBTAC in hydrogels. Please explain why?
There is indeed an interesting variation of the bactericidal effect, especially for the series with commercial chitosan and chitosan obtained from commercial chitin. Although it was expected to have an increasing bactericidal effect for all the samples with higher amounts of VBTAC, like the series with chitosan prepared from shrimp shells, we believe and concluded that the bactericidal effect is also affected by the more compact structure of hydrogels, as indicated by the swelling degrees. However, the increased bacterial retention of the sample with high amount of VBTAC, and actually of all the samples in the series with chitosan prepared from shrimp shells may also be influenced by the content of minerals present in the chitosan structure. Since this was just an observation, we only added the following explanation: “Additional to the SD, the significant reduction capacity for bacteria registered for the samples in the series of semi-IPNs based on SHC may also be linked to the content of native minerals present in the chitosan structure.”
- More discussion could be provided. For example, why several types of chitosan were selected to conducted this study?
And why the SHC-IPN sample exhibited better bactericidal effectiveness?
Several types of chitosan were selected for this study to underline the difference between the commercial types and the one extracted from shrimp shells without demineralization, sustaining in this way the cost-effective ecological applications and environmental engineering, namely wastewater treatment. The reasoning was provided in the introduction section, as follows: “The reason for using mineral enriched-chitosan (i.e., chitosan obtained by-passing the demineralization stage [52]) lies in the intention of improving the stability for chitosan-based hydrogels using the native minerals present in its structure (mainly calcium carbonate). Plus, the process of producing this type of chitosan and, thereafter, the semi-IPN hydrogels is also remarkably cost-effective and environmental-friendly, with regard to the carbon footprint of energy-consuming demineralization process [53]….. For demonstrating the performance of these new semi-IPN hydrogels based on miner-al-enriched-chitosan, the comparison experiments were performed using commercial chitosan and chitosan obtained from commercial chitin.”
The semi-IPNs with SHC exhibited better integrity during the swelling experiments and increased bactericidal effectiveness, compared to their correspondents in the series with commercial chitosan and chitosan prepared from commercial chitin, which may also be linked with the presence native minerals from the shells (e.g. Ca, Mg) indicating a competitive and promising potential for ecological purposes. This explanation was provided in Section 3.5.” “Additional to the SD, the significant reduction capacity for bacteria registered for the samples in the series of semi-IPNs based on SHC may also be linked to the content of native minerals present in the chitosan structure.”
- In the introduction section, several water treatment methods have been mentioned. Besides, solar steam generation is also an emerging method used for clean water production in recent years. The recent published works may be added, such as Adv. Mater. 2023, 35, 2209015,https://doi.org/10.1016/j.scib.2023.01.017
The authors would like to thank the reviewer for the suggestion and mention that the indicated work was added as reference [9].
- The hydrogels prepared in this work showed effective bactericidal effect for wastewater treatment. Recently, it was also reported that such antibacterial property of polymeric materials could facilitate the stable and durable clean water remediation performance (Materials Horizons, 2023,10, 268-276). Therefore, the bactericidal benefit of this hydrogel consisting of chitosan and ammonium-quaternary hydrogels may be also mentioned in the discussion. The clean water remediation performance and the bactericidal benefit of the obtained hydrogel consisting of chitosan and ammonium-quaternary were thoroughly discussed in the Introduction section. In this regard, the mentioned reference was added in the Manuscript as reference [10].
Round 2
Reviewer 2 Report
The revised version of the manuscript could be accepted for publication.
Reviewer 3 Report
No other comments.